# Effect of Agavins and Agave Syrup Use in the Formulation of a Synbiotic Gelatin Gummy with Microcapsules of *Saccharomyces Boulardii*

**DOI:** 10.3390/gels10050299

**Published:** 2024-04-26

**Authors:** Liliana K. Vigil-Cuate, Sandra V. Avila-Reyes, Brenda H. Camacho-Díaz, Humberto Hernández-Sánchez, Perla Osorio-Díaz, Antonio R. Jiménez-Aparicio, Paz Robert, Martha L. Arenas-Ocampo

**Affiliations:** 1Instituto Politécnico Nacional-CEPROBI, Carretera Yautepec-Jojutla, Km.6 calle CEPROBI No.8, Colonia San Isidro, Yautepec C.P. 62730, Mexico; kvigilc1800@alumno.ipn.mx (L.K.V.-C.); bcamacho@ipn.mx (B.H.C.-D.); posorio@ipn.mx (P.O.-D.); arjaparicio@gmail.com (A.R.J.-A.); 2CONAHCyT-Instituto Politécnico Nacional-CEPROBI, Carretera Yautepec-Jojutla, Km.6 calle CEPROBI No.8, Colonia San Isidro, Yautepec C.P. 62730, Mexico; 3Departamento de Ingeniería Bioquímica, Escuela Nacional de Ciencias Biológicas, Instituto Politécnico Nacional, Unidad Profesional Adolfo López Mateos, Mexico City C.P. 07738, Mexico; hhernan1955@yahoo.com; 4Departamento de Ciencias de los Alimentos y Tecnología Química, Facultad de Ciencias Químicas y Farmacéuticas, Universidad de Chile, Santos Dumont 964, Independencia, Santiago C.P. 8380494, Chile; proberts@uchile.cl

**Keywords:** synbiotic gummies, gelatin network, agave fructans, image digital analysis

## Abstract

Agavins are reserve carbohydrates found in agave plants; they present texture-modifying properties and prebiotic capacity by increasing the viability of the intestinal microbiota. Through its hydrolysis, agave syrup (AS) can be obtained and can be used as a sweetener in food matrices. The objective of this work was to evaluate the effect of the variation in the content of agavins and AS on the physical, structural, and viability properties of *Saccharomyces boulardii* encapsulates incorporated into gelatin gummies. An RSM was used to obtain an optimized formulation of gelatin gummies. The properties of the gel in the gummy were characterized by a texture profile analysis and Aw. The humidity and sugar content were determined. A sucrose gummy was used as a control for the variable ranges. Alginate microcapsules containing *S. boulardii* were added to the optimized gummy formulation to obtain a synbiotic gummy. The viability of *S. boulardii* and changes in the structure of the alginate gel of the microcapsules in the synbiotic gummy were evaluated for 24 days by image digital analysis (IDA). The agavins and agave syrup significantly affected the texture properties (<1 N) and the Aw (>0.85). The IDA showed a change in the gel network and an increase in viability by confocal microscopy from day 18. The number of pores in the gel increased, but their size decreased with an increase in the number of *S. boulardii* cells. Agavins and cells alter the structure of capsules in gummies without affecting their viability.

## 1. Introduction

Despite the impact of the contingency caused by COVID-19 in 2020, in 2022, the candy market in Mexico experienced an increase in consumption, representing an income of MXN 27.6 billion. Gummy candies have the greatest participation in the value and volume of the confectionery industry: 24.5% and 35.8%, respectively [1]. In 2008, a per capita sweets consumption of 4.5 kg was recorded in Mexico. In 2022, the export of sweets increased by 25.7% compared with 2021 [1,2]. Mexico has the second-highest consumption in Latin America and is ranked first as an exporter of sweets worldwide. The WHO [3] recommends that sugar consumption should not exceed 10% of caloric intake per day, so the use of natural sweeteners, like agave syrup, is recommended [4,5]. Sweets can be used as vehicles for bioactive compounds and administered to children or the elderly, providing a benefit to health with a functional character [6].

Prebiotics and probiotics have gained popularity due to their ability to beneficially modify the intestinal microbiota [7,8]. The most up-to-date definition of a prebiotic was given by the International Scientific Association for Probiotics and Prebiotics (ISAPP), which defines a prebiotic as “a substrate that is selectively used by host microorganisms, conferring a health benefit” [9]. Agavins (agave fructans) are low-molecular-weight carbohydrates structured by fructose chains that have branches in the β-(2-1) or β-(2-6) bonds [10,11] and a degree of polymerization (DP) ranging from 4 to 10 units in *Agave angustifolia* Haw. fructans, but they also have been found ranging from 2 to 60 units in *Agave angustifolia* Haw. fructan fractions [12,13]. In the colon, they are fermented by the intestinal microbiota, generating short-chain fatty acids [14,15]. Technologically, fructans have applications in a wide variety of foods, such as sugar substitutes, fat substitutes, texture improvers, and stabilizers, among others [16,17]. The interaction of polysaccharides with proteins like gelatin generates patterns that are spatially heterogeneous, directly influencing the optical, thermal, and mechanical properties of the materials related to flavor or appearance [18].

Probiotics are “live microorganisms that, when administered in adequate quantities, confer a health benefit to the host”, and this allows for a healthy intestinal microbiota [19,20,21]. *Saccharomyces boulardii* is a probiotic yeast; it has been shown to have resistance to most prescribed antibiotics, preventing diarrhea caused by their consumption [22]. Its growth temperature is the body temperature of 37 °C, and it presents high viability, conditions that make it an option for incorporation into food matrices [23]. To facilitate its addition and maintain or increase viability during its shelf life, encapsulation with prebiotic materials can strengthen the availability and stability of the microorganism, giving rise to a synbiotic product whose main objective is to enhance the beneficial effect, creating synergism between the prebiotic and the probiotic through their interaction [20,24]. The encapsulation of *S. boulardii* with sodium alginate and agavins has proven to be a good option. Additionally, agavins, in addition to protecting *S. boulardii* during the gastrointestinal process, have also been shown to retain their prebiotic effect [13,25].

Additionally, with tools such as digital image analysis (IDA) in micrographs of alginate microcapsules with agavins that contained *S. boulardii*, a network established by these materials within the microcapsule called beads internal networking (BIN) or mesostructure has been observed; it is established mainly by the agavins distributed in the microcapsule, and in which the distribution of the pores and the intersections of the network in the gel are measured [25,26]. In works in which synbiotic microcapsules with algae polysaccharides have been incorporated into gummy-type sweets, a double encapsulation of probiotics such as *Lactobacillus rhamnosus* has been simulated [27]. However, the use of gelatin as a gelling agent for synbiotic gummies has been observed to have better general acceptability than the use of agar [28]. Likewise, the use of sweeteners such as commercial agave syrup to reduce the use of sugar in gelatin gummies has been shown not to significantly affect the texture properties of the candy. Therefore, it can be incorporated into new gelatin gummy formulations [29,30]. The use of agave syrup in combination with agavins or alone has shown a prebiotic effect in gels [31,32].

Therefore, the aim of this work was to evaluate the effect of the incorporation of agavins and agave syrup from the species *Agave angustifolia* Haw., encapsulating *Saccharomyces boulardii* in the design, and to characterize a synbiotic gelatin gummy.

## 2. Results and Discussion

### 2.1. Formulation of a Gummy with Agavins and Agave Syrup

The gel texture of gummies can be affected by the concentration of the gelling agent, sweeteners, and type of sugar. In this work, the amount of gelling agent was not modified, so the results in the modification of the texture of the gels with a rest time of 24 h at 25 °C (gummies) were directly related to the addition of the type of syrup and sugar used. The addition of soluble solids, such as sucrose or glucose syrup, to aqueous solutions stabilizes the folded structure of globular and fibrous proteins due to their ability to develop hydrogen bond-type structures [33] like the one that is formed in the gelatin network in water, modifying its gelation properties, making the gel more rigid, and decreasing its deformation because of the increase in the melting point due to the increase in the stability of the triple helical chain. Therefore, it is a challenge to completely replace the use of sucrose in the production of sweets, such as gummies [30,32]. In this work, sucrose was replaced by agavins, and corn syrup was replaced by agave syrup in the formulation of gelatin gummies.

Predictive equations describe the effect of the agavin content (x) and agave syrup content (y) on the hardness, cohesiveness, adhesiveness, elasticity, gumminess, and Aw of gelatin gummies. The information is shown in more detail in Equations (1)–(6).
Hardness = 1318.22 − 29.38*x + 0.21*x^2^ − 24.09*y + 0.17*y^2^ + 0.18*x*y + 0(1)
Cohesiveness = −0.65 + 0.074*x − 0.00087*x^2^ + 0.016*y − 0.000065*y^2^ − 0.00023*x*y + 0(2)
Adhesiveness = 1.34 − 0.031*x + 0.00054*x^2^ − 0.033*y + 0.00065*y^2^ − 0.00035*x*y + 0(3)
Elasticity = 52.98 − 0.20*x + 0.0012*x^2^ + 1.49*y − 0.022*y^2^ + 0.0027*x*y + 0(4)
Gumminess = 1227.88 − 25.55*x + 0.17*x^2^ − 23.05*y + 0.17*y^2^ + 0.17*x*y + 0(5)
Aw = −0.0038 + 0.015*x − 0.000074*x^2^ + 0.029*y − 0.00025*y^2^ − 0.00024*x*y + 0(6)

Gelatin gummies with agavins had greater softness (0.21 N to 3.16 N) compared with a commercial gummy (CG), which had a hardness of 11.64 ± 1.12 N. The same effect was observed for the gumminess. The gelatin gummies formulated with agavins presented a gumminess of 0.20 N to 3.20 N, values lower than that shown by CG, with 11.08 ± 0.8 N. This indicates that less force is required to disintegrate the gummies before being swallowed [34]. On the other hand, the control gummy (A40) made with sucrose presented a hardness value of 0.84 ± 0.09 N and a gumminess of 0.82 ± 0.09 N, values like those obtained for the gelatin gummies with agavins and agave syrup.

Figure 1 shows the Pareto chart of standardized effects of the ANOVA (α = 0.05), in which a significant effect is observed in all response variables. Nevertheless, the content of agave syrup and agavins, as well as the interaction of both, had a significant effect on Aw; therefore, formulations with high contents of agave syrup and/or agavins had a higher Aw, as shown in the response surface graph. Periche et al. (2014) [35] reported similar results in gelatin gummies formulated with isomaltose and fructose, which presented Aw values of 0.7210 to 0.9080, where the treatments consisting of fructose and glucose syrup showed the highest Aw.

The hardness and gumminess values obtained for CG were high, probably due to the long storage time during shelf exposure, as well as the low Aw, in which a value of 0.6654 ± 0.0113 was obtained. Wang et al. (2024a) [36], observed that when there was a reduction in water, there was greater hardness. Typically, gummies are characterized by having an Aw between 0.5 and 0.75; however, when they have low water activity and are not stored correctly, a substantial increase in hardness is promoted. This could also be because when there is a higher solids content due to the glucose and sucrose content in commercial gummy formulations, water availability is lower.

The Aw decreases because of the drying of the gummy during the long storage time on the shelf, where other types of phenomena can also occur, such as Maillard reactions between sugars and gelatin [37,38]. On the contrary, a sweet with a high Aw promotes greater softness [33], as observed in gummies formulated with agavins and agave syrup, which presented Aw values between 0.7524 to 0.8540, as well as in A40, which presented an Aw of 0.7298 ± 0.0318. Agavins are made up of linear and branched fructose chains, while agave syrup, on the other hand, is made up of free and linear fructose units due to hydrolysis. It is common to use sucrose and glucose syrup as conventional sugars in soft candy formulations; however, glucose syrups are obtained mainly from the hydrolysis of starch, so the degree of hydrolysis can vary, having variable percentages of sugars like glucose and maltose, oligomers (3–9 degrees of polymerization, DP), and residual polymers (≥10 DP) [5,33]. The gummies are between 70% and 80% carbohydrates, decreasing the water content to 16–20%. The syrup usually covers 50% or more of the total sweetening solids, with polymers with a DP of ≥10 in the mixture with the gelatin, influencing the structure and property relationship of soft sweets such as gummies [36].

Wang et al. (2024b) [39] thoroughly investigated the phase separation phenomena of gelatin–glucose syrup mixtures and observed molecular incompatibility between gelatin and polysaccharides with a DP of >10 in glucose syrup (GS). In agave syrups, the composition, in which fructooligosaccharides with a DP of <10 predominate, in addition to residual fructose, gives them the characteristic of having high hygroscopicity compared with sucrose or glucose, directly affecting the gel network [38].

The cohesiveness results indicated that the use of agavins and agave syrup in the gelatin gummies improved this property. In most gummies with agavins, this property was slightly higher compared with the CG (0.95) and A40 (0.98). It may be due to the fact that the chains of fructose units that constitute the agavins have a large number of hydroxyl groups, which could contribute to the formation of structures with high cohesiveness in the gelatin gummy gel network [33,38]. With the use of inulin (a linear chain fructan), some authors have indicated that a high DP can form a three-dimensional network that reinforces the gelatin network through hydrogen bonds [32].

Adhesiveness in gummies is related to how sticky a candy is; however, to reduce this undesirable characteristic, natural coatings such as carnauba wax or vegetable oil are used in such a way that the ingredients or their interaction do not have a significant effect on this parameter [35], and therefore, these results are not shown. Finally, the elasticity results (71.92 mm to 75.6 mm) of the gelatin gummies with agavins were similar to those obtained for the CG (72.38 ± 0.3 mm) and A40 (75.03 ± 0.06 mm). In this way, when replacing sucrose and glucose syrup, Aw was the most affected factor. However, it was observed that an increase in the amount of agave syrup with the same proportion of agavins presented a greater number of solids present in the matrix, so the water activity decreased, but it also decreased the hardness and gumminess, which is not desirable. Although there was high water activity at the midpoint of the treatments, an increase in agavins had a significant effect on cohesiveness, while elasticity was mainly affected by agave syrup. These functional ingredients altered and improved the textural properties of the gelatin gummies, in line with what other authors have reported [29,31], in which the ingredients form a more uniform network because due to the nature of the branched structure of the fructooligosaccharides and DP of the agavins, they form stronger internal bonds capable of supporting secondary deformations [40] despite having high Aw. The predicted values and observed values of TPA and Aw are shown in Table 1. The optimum value was of 0.90 for a formulation with 40.68 g of agavins and 40.68 mL of agave syrup, similar to the formulation of SA40. The observed values of SA40 were similar to the predicted values shown in Table 1. Microcapsules were added to this formulation to obtain the synbiotic gummy.

#### Optimization of the Formulation of the Gummy

The desirability specifications to obtain the optimal gummy formulation and the desirability surface graph to locate the simultaneous optimum are shown in Table 2 and Figure 2. It was observed that most of the combinations of agavins and agave syrup studied were very close to the optimal point (desirability = 1); however, formulations with a low content of agavins and agave syrup were far from it. Finally, the formulation in the central point, S4A4, obtained the numerical optimum of desirability in the analysis, with a global desirability of 0.90. The formulation was made up of 40.68 g of agavins and 40.68 mL of agave syrup. This formulation was used to add the *S. boulardii* microcapsules.

### 2.2. Obtaining Synbiotic Gummies

#### 2.2.1. Physicochemical Characterization of Synbiotic Gummies

Table 3 shows the results of the physicochemical characterization of the synbiotic gelatin gummy (SIG) in comparison with the optimal gelatin gummy (S4A4), in which the microcapsules were not yet included, and the CG. The results of Aw and moisture content showed a significant decrease compared with S4A4 and the CG. The Aw present in the SIG is ideal to ensure the microbiological stability of the gummy, in accordance with the conditions described in Section 4.3.1 [33].

By replacing sucrose and glucose syrup in the gelatin gummies formulation with agavins and agave syrup, synbiotic gelatin gummies (SIGs) were obtained with a total sugar content of 28.03 ± 3.35%, significantly lower than that reported for the CG (51.5%) [41]. However, they cannot be considered synbiotic gelatin gummies with reduced sugar because according to NOM-086-SSA1-1994, in order to be considered reduced-sugar gummies, they should have half the total sugar content reported for the commercial product. Significantly fewer reducing sugars were obtained in the SIGs compared with the CG, meaning they have less glucose and fructose content. On the other hand, in non-reducing sugars, which refer to agavins, a significantly higher content was obtained in the SIG than in the CG, demonstrating that the predominant types of sugars in the SIG are agavins (prebiotic). The texture profile analysis (TPA) results indicated that the addition of microcapsules did not significantly modify the texture properties of the SIG.

#### 2.2.2. Viability of Microencapsulated Saccharomyces Boulardii

The cell count of *S. boulardii* in the culture medium before encapsulation was 7.5 × 10^8^ CFU/mL (8.4 ± 0.16 log CFU/mL). After encapsulation, a cell count of 6.3 × 10^7^ CFU/g of capsules (7.1 ± 0.21 log CFU/g) was recorded. Viability loss of 1.1 log cycles (CL) was observed, which is normal with this encapsulation method. Some authors have recommended the addition or use of a prebiotic as a wall material during the encapsulation process that helps maintain the viability of the probiotic microorganism during the storage stage [24,25]. Zamora-Vega et al. (2012) [24] obtained *S. boulardii* microcapsules using inulin and cactus mucilage as a wall material, and a viability of 7.31 ± 0.31 log CFU/g of microcapsules was maintained, even after 30 days at 4 °C. They reported that inulin did not lose its prebiotic function. On the other hand, Chávez Falcon et al. (2022) [25] observed that agavins improved the polymer network of the gel, providing a heterogeneous internal structure in the microencapsulation of *S. boulardii* by ionic gelation; in addition, the use of 5% agavins increased the viability during encapsulation.

The viability of *S. boulardii* in the gummies reached 6.7 × 10^8^ CFU/g or 8.8 log CFU/g of gummies. The FAO/WHO (2002) [19] definition considers that a food that is considered a probiotic must have a viability in the range of 1 × 10^7^ to 1 × 10^9^ CFU/mL of the microorganism at the time of consumption. Therefore, according to this definition, it can be considered that this requirement was met, obtaining gummies that contained probiotics and prebiotics and, finally, a synbiotic product. Lele et al. (2018) [28] developed synbiotic gummies with apple pulp and psyllium dietary fiber using the strains of *Lactobacillus plantarum* LUHS135 and *Lactobacillus paracasei* LUHS244 as probiotics; the gummies presented a viability of 6.4 log_10_ CFU/g and 6.5 log_10_ CFU/g, respectively. In this way, it is proven that the microencapsulation of the probiotic is important to maintain high viability once it has been incorporated into the food, especially when added to foods such as gummies, which require a thermal process for their preparation. The stability of the viability of the *S. boulardii* microencapsulated within the synbiotic gelatin gummies was measured during almost a month of storage, and colony-forming units were obtained until day 14 of plate culture (7.04 log CFU *S. boulardii*/g gummies) (Figure 3). Subsequently, there was no growth on the plate (Day 21 = 0 log CFU *S. boulardii*/g gummies).

### 2.3. Morphometric Characterization of Synbiotic Gummies

Through the seeding method on agar plates, it was not possible to obtain the viable count; however, the viable cells were detected through laser scanning confocal microscopy (LSCM) (green cells, Figure 3), from the beginning until the end of the experiment (day 24 = 94% viability of *S. boulardii*). This behavior is like that observed in some microorganisms when exposed to stress; they remain metabolically active but in a dormant or non-culturable state and are called viable but non-culturable (VBNC) [42]. One of the questions that arises for this work after the MCBL images (Figure 3) is why *S. boulardii* increases its population density inside the gelatin gummies rather than only stabilizing in the initial population or decreasing, as seems to happen with the viable count on agar plates (Figure 2), upon entering a VBNC state. Yang et al. (2022) [43] provoked the VBNC state in *Saccharomyces cervisiae* using cold wort with isomerized hops and found that this state is promoted by the tricarboxylic acid (TCA) cycle, the ABC transporter, organic acid metabolism, and oxidoreductase activity. Likewise, stress response proteins, elongation factors, ribosomal proteins, kinase transporters, and fluoride export have been identified in *S. boulardii*, which could help its adaptation under stress conditions [44,45].

In a principal component analysis (Figure 4) of the interaction variables of the variables determined with the image analysis and cell counts, it was shown that the first component is made up of the structure variables of the reticulation network of the microcapsules (entropy, gel intersections, triple points, quadruple points, and average number of pores), while the second is governed by the viability of *S. boulardii* (% viability, average length of branches, total cells, green and red, and size of pore). In this way, it was observed that the structure of the microcapsule gel was maintained until day 11; starting on day 14 when *S. boulardii* entered the VBNC state, the entropy in the gel network increased. The skeletonized images (Figure 5) show how the intersections and quadruple points of the network increase, while in the gap images, it is observed that the pores become smaller (blue and black points) and that their frequency increases (Figure 4). Chávez Falcon et al. (2022) [25] observed that agavins are the material that confers the complexity of the network in the microstructure of the microcapsule. However, in this work, it was observed that when these microcapsules were added to the gel to form the synbiotic gummy, the network became more complex, not only because of the difference in materials (alginate, gelatin, sugars, and additives) but also because as the days passed, the complexity increased due to the increase in *S. boulardii* cells inside, as mentioned above.

On days 14 and 21, the viability of *S. boulardii* and the number of total cells increased, as observed in the LSCM micrographs of the capsules (Figure 3) where fluorescence was observed throughout the microcapsule, but later, when viability decreased on days 18 and 24, there was only fluorescence in the periphery. This image provides an idea that indicates that *S. boulardii* can migrate toward the synbiotic gelatin gummy where it can use the agavins, fructose, glucose, and gelatin in the matrix as a carbon source. In Figure 3, a micrograph taken of the synbiotic gummy, the interface between the gummy and the microcapsule is observed; a smooth structure of the gummy can be distinguished surrounding the microcapsule, where *S. boulardii* is observed (purple) to concentrate or migrate toward the periphery of the gum. This behavior could indicate that the reason why *S. boulardii* enters the VBNC state in the microcapsules contained in the synbiotic gummies is due to a lack of nutrients as well as oxygen [46,47]. Inside the gelatin gummy, there are bubbles, which decrease in size as the population density of *S. boulardii* increases, as seen in the MCBL images in Figure 3. That is, in the first 14 days, it consumes some of the carbon sources of the microcapsule and can then begin to migrate toward the gummy, where there is a greater quantity and diversity of nutrients, as well as oxygen, while those that remain in the microcapsule continue to consume the remaining nutrients, such as shorter chains of carbohydrates and dead cells, making use of the oxygen that the bubbles of the gelatin and alginate gel can in turn provide. A study on synbiotic candy that contained probiotic microcapsules showed that in the candy without a prebiotic ingredient, there was no viability of the probiotic before 14 days of storage, while the synbiotic gummy still presented viability [27].

Gummies are gels that have already had a rest time and in which a gelation process has already occurred. Wang et al. (2024a) [36] focused their work on the effects of water content on the gelling behaviors of gelatin–GS mixtures and explored gel properties upon cooling to better understand the molecular interactions and microstructure property relationships during the mixing stage. The microstructures formed at high temperatures are trapped within gel networks following rapid cooling.

## 3. Conclusions

The present study showed the influence of using agavins and agave syrup instead of sucrose and corn syrup on the viability and microstructure of the gel network formed between the interaction of alginate microcapsules and a gummy-type gelatin matrix. The agavins and agave syrup helped stabilize the gelatin network, presenting greater cohesiveness. The hygroscopicity of agavins allowed greater binding of water molecules in the gelatin network, increasing Aw and decreasing hardness, obtaining a soft and easy-to-swallow matrix. The addition of microcapsules did not modify these properties; however, changes were observed in the alginate/agavin network of the microcapsules inside the gelatin gummies. The increase in viable *Saccharomyces boulardii* cells resulted in a porous network of microcapsules, and these pores decreased in size over time during storage. The addition of agavins improved the survival of the probiotic yeast. The plate count did not show CFUs after several weeks; however, laser scanning confocal microscopy did show the metabolic activity of *S. boulardii* in the microcapsules. The above shows the presence of cellular stress, causing a viable but non-culturable state of the yeast in the gummies.

## 4. Materials and Methods

### 4.1. Material

To manufacture the gummies, commercial gelatin (290 Bloom, Coloidales Duche S.A. de C.V., Mexico), artificial strawberry flavoring (DEIMAN S.A. de C.V.), strawberry red coloring 240 (DEIMAN S.A. de C.V.), and food-grade citric acid (Farmacia París, S.A. de C.V., Mexico) were used. The fructans of *A. angustifolia* Haw. (agavins) (with a degree of polymerization (DP) ranging from 4 to 10 units) and agave syrup were obtained through a patented process in CEPROBI-IPN (patent 380041). Classic commercial gummies from the Ricolino brand were purchased from a local distribution store. As a probiotic microorganism, a commercial strain of *S. boulardii* (CNCM I-745, BIOCODEX, FLORATIL) was used in this study.

### 4.2. Experimental Design, Formulation and Optimization

For the formulation of the gelatin gummies, a central composite design was applied using STATISTICA 7.0 software (TIBCO Software Inc., Palo Alto, CA, USA). The variations in agavins (30 to 50 g) and agave syrup (30 to 50 mL) content were established as independent variables. The amount of gelling agent and citric acid was kept constant. The dependent variables were hardness (N), cohesiveness, elasticity (mm), adhesiveness (mJ), gumminess (N), and Aw. Eight formulations and two more central points were obtained, and the experimental conditions for the gummy formulation are shown in Table 4. Each treatment was carried out in triplicate.

On the other hand, a response surface methodology (RSM) design was used to determine the optimal conditions for each independent variable (agavin content and agave syrup content), considering the linear and quadratic form and interaction of the independent variables at a confidence level of 95% (*p* < 0.05).

The data for each variable were adjusted to a second-order regression model, as shown in Equation (7). The response surface methodology (RSM) was used to determine the optimal conditions for each independent variable (agavin content and agave syrup content), considering the linear, quadratic, and interaction form of the independent variables at a confidence level of 95% (*p* < 0.05):(7)$$Y=b_{0}+\sum_{1=1}^{2}b_{i}X_{i}+\sum_{1=1}^{2}b_{ii}X_{i}^{2}+\sum_{i=1}^{1}\sum_{j=i+1}^{2}b_{ij}+X_{I}+X_{J}$$
where *Y* is the estimated response for each variable. The superscripts *i* and *j* show the number of variables (*n = 2*); *b*_0_ is the intercept, *b_i_* is the linear coefficient, *b_ii_* is the quadratic coefficient, and *b_ij_* is the coefficient of interaction. Finally, *X_i_* and *X_j_* are the levels of each independent variable.

The optimization for the preparation of the gummies was carried out using the desirability function (D) [48]. The desirability specifications of the TPA (hardness, gumminess, cohesiveness, elasticity, and adhesiveness) and Aw variables were determined using as a reference the characteristics of a gelatin gummy sweetened with sucrose and corn syrup (A40). The other variable specifications were determined using a commercial gelatin gummy (CG), using 1 as the desirable value and 0 as the non-desirable value, as shown in Table 4. A high value of the exponents s and t was used when a high value was close to the objective value; on the other hand, a small value of s and t was used for accepting every value within the minor and major desirability. When s and t are 1, there is a linear increase for the desirability to the objective value [49,50].

### 4.3. Elaboration of Gelatin Gummies

The gummies were prepared according to the methodology in [35] with some modifications, replacing corn syrup and sucrose with agave syrup and agavins from *A. angustifolia* Haw., respectively. First, 0.5 g of citric acid was dissolved in 2.5 mL of hot distilled water; separately, 10 g of gelatin as a gelling agent was hydrated in 25 mL of water and allowed to rest for 30 min. According to the treatments obtained in the experimental design (Table 4), the amounts of agavins (variable 1) and agave syrup (variable 2), were mixed together in 15 mL of water. The mixture was heated to 90 ± 5 °C with constant stirring for 10 min. After this time, the hydrated gelatin, 0.5 mL of artificial strawberry flavoring, 0.2 g of strawberry red 240 coloring, and the dissolved citric acid were added. The mixture was placed in a water bath for 30 min; once this time passed, the mixture was poured into silicone molds previously greased with vegetable oil and allowed to cool for 90 min at room temperature. The gummies were unmolded and stored at room temperature (25 °C) in sterile resealable bags for later analysis to avoid moisture loss. An incubator was used at a temperature of 25 °C (LAB-LINE, Model R3525, Melrose, IL, USA). As a comparison method for the type of sugar and sweetener, a control gummy (labeled A40) was made in a traditional way; sucrose and corn syrup were used, and the same procedure previously described was followed.

#### 4.3.1. Determination of Texture and Physicochemical Properties

To see the effect that the type of sugar and the amount of added syrup had on the gelatin mixtures with different proportions of agavins and agave syrup, an analysis of the texture properties and the Aw in the solidified gels was performed in a control gummy (A40), treatments (see Table 3), and synbiotic gummies.

The TPA was performed on gel cubes (height: 1 cm, total dimensions: 1 cm^3^) with 24 h of rest at room temperature to ensure hardening of the gels using a texture analyzer (TAXT2, Stable Micro Systems Ltd., Caerphilly, Gales, UK). The profile consisted of two consecutive cycles of 40% compression at a speed of 1 mm/s; a 4.5 kg load cell and a 12 mm diameter cylindrical probe were used [35,49]. Five samples were used in triplicate for each treatment. Parameters of hardness, cohesiveness, elasticity, adhesiveness, and gumminess were obtained in this test.

The Aw of the gelatin gummies was determined using an Aqua Lab 4TE (Decagon Devices Inc., Pullman, WA, USA). Measurements were performed in triplicate at an ambient temperature of 25 °C.

### 4.4. Preparation of Synbiotic Gummies

To the optimized gelatin gummy formulation (S4A4) with agavins and agave syrup, microcapsules with *S. boulardii* were added to obtain synbiotic gummies. The methodology for obtaining these microcapsules is described as follows.

#### 4.4.1. Growth Conditions of *Saccharomyces boulardii*

The commercial strain of *S. boulardii* (CNCMI-745, BIOCODEX, FLORATIL, Beauvais, France) was reactivated by adding the contents of a capsule to 100 mL of YPD broth (yeast peptone dextrose) (1% glucose, 1% casein peptone, 1% yeast extract, and NaCl 0.5 g/L) with shaking at 200 rpm and at a temperature of 37 °C for 12 h (LAB-LINE; Incubator shaker; Orbit; Model R3525, Melrose, IL, USA). *S. boulardii* was counted by the microdrop plate culture method [50]. A solid YPD medium was used, incubated at 37 °C for 24 h, and plate counting was performed (CFU/ mL) [25].

#### 4.4.2. Microencapsulation

*S. boulardii* cell concentrates inoculated at 5% were prepared in 100 mL of YPD broth with shaking at 200 rpm and at a temperature of 37 °C for 12 h (LAB-LINE; Incubator shaker; Orbit; Model R3525, Melrose, IL, USA). Subsequently, the cell button was collected in sterile 50 mL centrifuge tubes and centrifuged at 10,000× *g* at a temperature of 4 °C for 10 min. The cells were washed twice with a sterile PBS solution. By decanting, the supernatant was removed to obtain the button-shaped cell concentrate. The cell button from both tubes was concentrated into a single tube.

Encapsulation was carried out using the ionic gelation method under sterile conditions in a laminar flow hood. The cell bud was added to a solution of 100 mL of 1% sodium alginate (REASOL^MR^, molecular weight 216 g/mol, purity 95–100%, Mexico City, Mexico) in sterile water and 5% agavins (sterilized with 4 kGy gamma rays; ICN-UNAM). The solution was placed in a 5 mL syringe (21G gauge × 32 mm needle). The microcapsules were obtained by dripping into a 0.2 M CaCl_2_ solution, where they were kept for 30 min to harden and strengthen cross-linking. Finally, the microcapsules were collected on Whatman No. 4 filter paper [21] and subsequently stored at 4 °C, keeping them in a 0.9% saline solution until use.

#### 4.4.3. Preparation of Synbiotic Gelatin Gummies

To prepare the synbiotic gelatin gummies, the methodology set out in Section 4.3 was followed; However, for the addition of the *S. boulardii* microcapsules, a layer of the gummy mixture was first poured into the molds. When the mixture reached a temperature of 50 ± 5 °C, 200 mg of the *S. boulardii* microcapsules was added, and it was allowed to cool for 20 min at 4 °C. After this time, another layer of the gummy mixture was added, and it was allowed to cool for 90 min at room temperature. The gummies were unmolded and stored in sterile resealable bags for sampling at room temperature for later analysis. An incubator was used at a temperature of 25 °C (LAB-LINE, Model R3525, Melrose, IL, USA).

### 4.5. Physicochemical Properties of Synbiotic Gelatin Gummies

The texture profile analysis and determination of Aw were carried out as mentioned in Section 4.4. The moisture content was determined [51,52]. The content of total sugars and reducing sugars of the synbiotic gummies was determined according to NOM-086-SSA1-1994 by means of a volumetric titration [53]. Non-reducing sugars were obtained by the difference in the content of total sugars and the content of reducing sugars.

### 4.6. Determination of Viability

The viability of *S. boulardii* in the samples of synbiotic gelatin gummies was determined during its storage at 25 °C for 4 weeks at 25 °C by plating on YPD agar plates and by laser scanning confocal microscopy (LSCM). To release *S. boulardii* from the alginate gel, a synbiotic gummy (2 g) was dissolved in 1% sodium citrate solution and plated once a week by the microdrop method on YPD agar plates [25,27,50].

In the viability analysis by LSCM, Zeiss LSM 800 (Carl Zeiss, Jena, Germany) equipment was used, coupled to an AxioCam HD color Model 305 (Carl Zeiss, Jena, Germany). The synbiotic gummies were dissolved in 2 mL of sterile phosphate buffer (PBS) for 15 min at 300 rpm and 37 °C. Once the microcapsules were released, they were stained with Acridine Orange (AO) and Propidium Iodide (IP) [54]. Samples were mounted on slides to monitor *S. boulardii* viability by fluorescence in MCBL at a wavelength of 488 nm at 4.5% excitation for NA (viable cells—green) and 561 nm at 2.0% for IP (non-viable cells—red). Smart SEM 2.6 Blue edition software (Carl Zeiss Microscopy, Cambridge, UK) was used, and 5 micrographs of the central area of each microcapsule were obtained. The micrographs were acquired in the “Tiles” mode, which consists of the formation of macrophotographs by joining images in the XY axis thanks to its motorized stage. A 5× and 20× apochromatic objective (Plan-Neofluar 20×/0.5) was used, with a numerical aperture of 0.8 and 1.3, respectively, stored in RGB in TIFF format at a resolution of 2048 × 2048 pixels.

### 4.7. Digital Image Analysis

#### 4.7.1. Viability Analysis

ImageJ v.1.54h software (National Institutes of Health, Bethesda, MD, USA) was used. The “color threshold” tool in the “Lab” color space was used to select and separate viable and non-viable cells and count them using “analyze particles”. The percentage of viability was calculated using Equation (8):(8)$$Viability\;\left(\%\right)=\left(\frac{Viable\;Cells}{Total\;Cells\;}\right)\times \left(100\right)$$
where the number of viable cells was differentiated as green cells, non-viable cells were differentiated as red cells, and total cells were calculated as the sum of red cells and green cells.

#### 4.7.2. Microstructure Analysis

The structure and behavior of the alginate gel network in the microcapsule in which *S. boulardii* was found inside the synbiotic gelatin gummy were determined. Digital image analysis was performed using ImageJ2 2.14.0/1.54f software (National Institutes of Health, Bethesda, MD, USA). Micrographs were binarized and skeletonized using the Skeletonize 2D/3D plugin. The microcapsule internal network (BIN) information, observed with skeletonization, was obtained with the GLCM (Gray Level Co-Occurrence Matrix) plugin and the “analyze skeleton” tool. Likewise, the Bone J “Local Thickness” plugin and the “histogram” tool were used to observe the mesostructure of the gel, modified by *S. boulardii* in the alginate microcapsules with agavins, inside the synbiotic gelatin gummy [25].

### 4.8. Statistical Analysis

Statistical analysis was performed using one-way analysis of variance (ANOVA). Significant differences between the means were compared using Tukey’s post hoc methodology (*p* < 0.05). Mean comparison analysis was performed using STATISTICA 7.0 software (TIBCO Software Inc., Palo Alto, CA, USA). All analyses were performed in triplicate (*n* = 3), and the results are expressed as the mean ± standard deviation (SD). To analyze the effect of *S. boulardii* viability and the structure of the microcapsules during storage, a principal component analysis by covariance was performed; Minitab V 18.1 statistical software (Minitab Inc., State College, PA, USA) was used.

## Figures and Tables

**Figure 1 gels-10-00299-f001:**
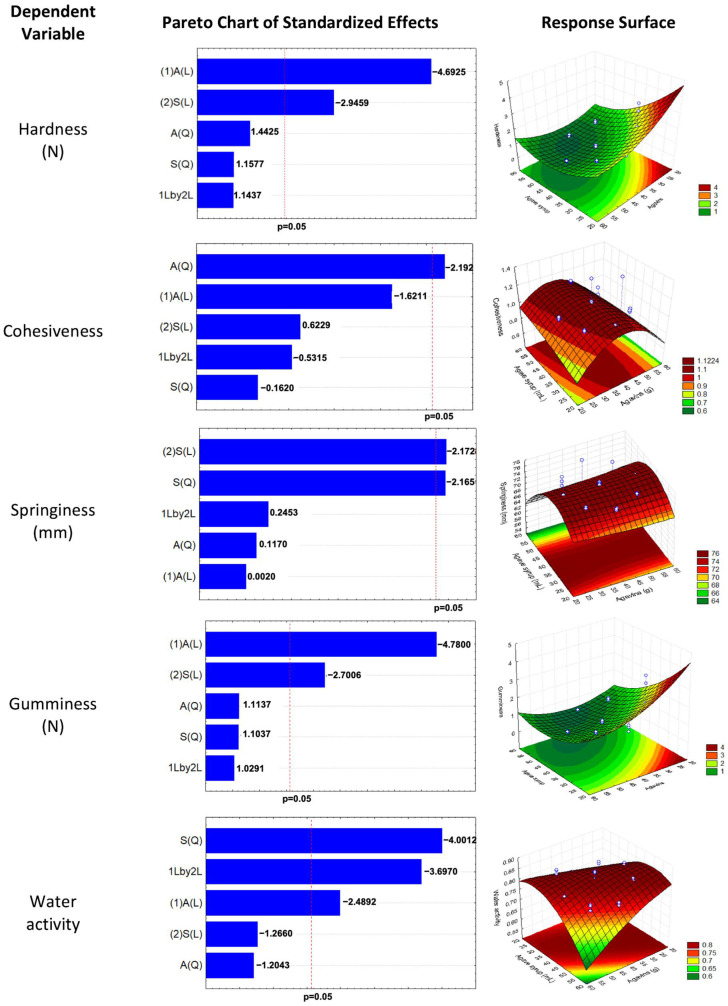
Minitab Pareto chart of the standardized effects of agave syrup and agavin content on the texture profile analysis and water activity of the gelled gummies with 24 h of rest. The red line indicates statistical significance (α = 0.05). A, agavins; S, agave syrup; L, Lineal; Q, Quadratic.

**Figure 2 gels-10-00299-f002:**
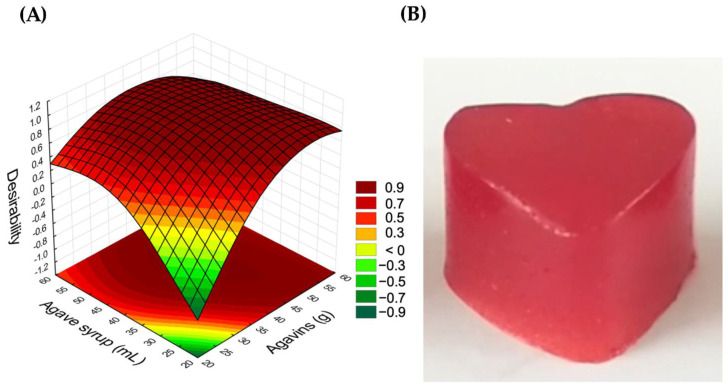
Optimal gummy formulation. (**A**) Desirability surface, (**B**) optimal gummy S4A4.

**Figure 3 gels-10-00299-f003:**
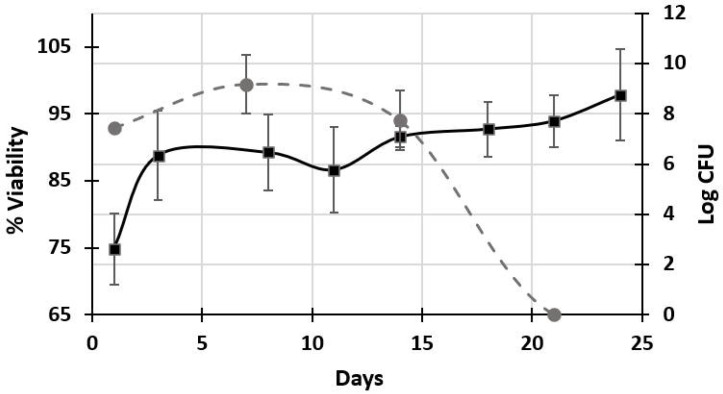
Viability of *S. boulardii* in the synbiotic gummy, during storage time at room temperature of 25 °C; (–●–) log CFU, (–⯀–) % viability.

**Figure 4 gels-10-00299-f004:**
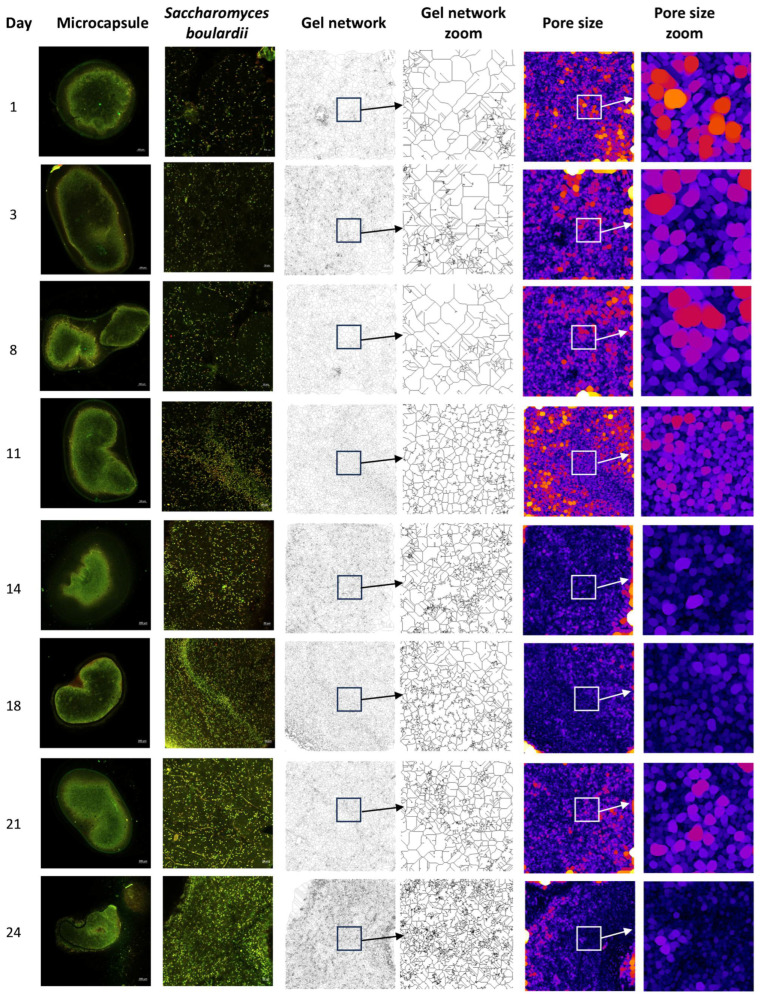
Stability of *S. boulardii* in the microcapsule gel inside of the synbiotic gummy.

**Figure 5 gels-10-00299-f005:**
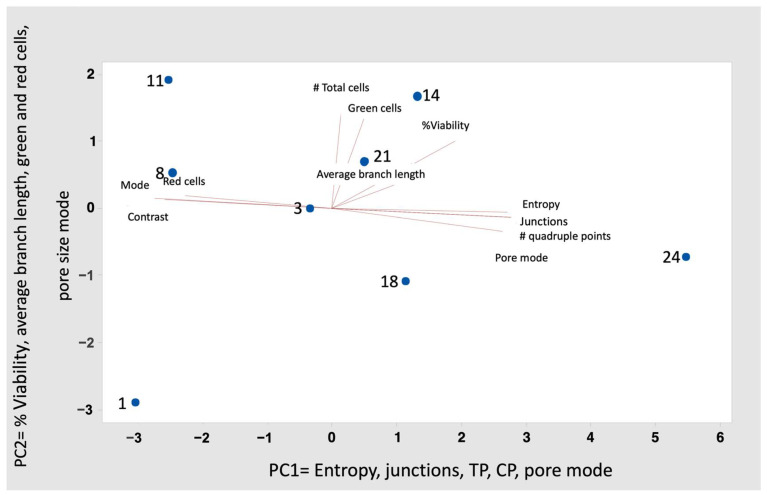
Principal components analysis of the gel network and viability parameters of *Saccharomyces boulardii* for 24 days in the synbiotic gummy.

**Table 1 gels-10-00299-t001:** Predicted value and observed value.

Dependent Variable	Predicted Value *	Observed Value *
Hardness (N)	0.82	0.78 ± 0.09
Cohesiveness	1.09	1.08 ± 0.13
Springiness (mm)	73.62	75.38 ± 0.42
Adhesiveness (mJ)	0.20	0.14 ± 0.03
Gumminess (N)	0.89	0.84 ± 0.08
Water activity	0.8193	0.8316 ± 0.023

***** *X* = 40, *Y* = 40, where: X = agavin content; Y = agave syrup content.

**Table 2 gels-10-00299-t002:** Desirability specifications to obtain the optimal gummy formulation.

Dependent Variable	Low Value	D	Medium Value	D	High Value	D	s	t
Hardness (N)	0.21	0.00	0.84	1.00	3.16	0.01	5.00	5.00
Cohesiveness	0.51	0.00	0.93	0.80	1.34	1.00	1.00	1.00
Springiness (mm)	54.70	0.00	65.40	0.50	76.10	1.00	1.00	1.00
Adhesiveness (mJ)	0.04	1.00	0.41	0.50	0.78	0.00	1.00	1.00
Gumminess (N)	0.20	0.00	0.84	1.00	3.20	0.01	5.00	5.00
Water activity	0.7524	1.00	0.8039	0.50	0.85	0.00	0.10	0.10

D, desirability value; s, s parameter; t; t parameter.

**Table 3 gels-10-00299-t003:** Physicochemical and texture characterization of the synbiotic gummy.

Property	SIG	S4A4	CG
Moisture content * (%)	32.45 ± 0. 37 ^a^	34.45 ± 0.11 ^b^	20–25 ** ^c^
Aw *	0.7465 ± 0.01 ^a^	0.8518 ± 0.004 ^b^	0.6572 ± 0.005 ^c^
Total sugars *	28.03 ± 3.35 ^a^	32.78 ± 2.74 ^a^	51.5 ** ^b^
Reducing sugars * (%)	11.76 ± 0.29 ^a^	12.13 ± 0.56 ^a^	51.1 ** ^b^
No reducing sugars * (%)	16.26 ± 3.26 ^a^	20.65 ± 3.15 ^a^	0.4 ** ^b^
Hardness (N)	1.73 ± 0.23 ^a^	0.79 ± 0.07 ^a^	11.64 ± 1.12 ^b^
Cohesiveness	1.02 ± 0.01 ^a^	1.07 ± 0.02 ^b^	0.95 ± 0.02 ^c^
Springiness (mm)	71.43 ± 3.68 ^a^	75.57 ± 0.5 ^a^	71.95 ± 4.24 ^a^
Gumminess (N)	1.32 ± 0.66 ^a^	0.85 ± 0.1 ^a^	11.08 ± 0.81 ^b^

* Values in physicochemical tests, *n* = 3; values in texture tests, *n* = 5, average ± SD, post hoc Tukey HSD test. Values with different letters in the same row showed significant differences (*p* < 0.05). SIG, synbiotic gummy; S4A4, optimized gelatin gummy formulation; CG, commercial gummy. ** Data obtained from references [32,40].

**Table 4 gels-10-00299-t004:** Experimental design for the gummy formulation.

Treatment	Agave Syrup (mL)	Agavins (g)	Treatment Key
1	30	30	S3A3
2	30	50	S3A5
3	50	30	S5A3
4	50	50	S5A5
5	40	40	S4A4 *
6	26	40	S26A4
7	54	40	S54A4
8	40	26	S4A26
9	40	54	S4A54
10	40	40	S4A4 *

* Central point. S, agave syrup; A, agavins.

## Data Availability

The data presented in this study are available on request from the corresponding author.

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
