# Peer review of "Effect of Agavins and Agave Syrup Use in the Formulation of a Synbiotic Gelatin Gummy with Microcapsules of *Saccharomyces Boulardii"

_gels, 2024, doi:10.3390/gels10050299_

Round 1

Reviewer 1 Report

Comments and Suggestions for Authors

The work presents the study and characterization of gummy candies using agavinas and agave syrup. The authors present an interesting work for the community in the development of symbiotics, however, the statistical rigor and description of the methodology should be expanded.

Introduction

The authors should include background information on the topic of the work. For example, López-Palestina et al 2023 ttps://doi.org/10.47836/ifrj.30.6.18  and Rodriguez-Rodriguez, R.; et al (2022). https:// doi.org/10.3390/molecules27154902  

Results.

The samples present very low textural properties, hardness and gumminess. The authors should make a comparison of these properties with similar products reported in literature.

The water activity of the gummy samples was very high. This can eventually lead to microbiological contamination problems, as well as a low shelf life of these products. What was the water activity of the commercial sample, in the desirability model, what was the criterion for selecting the water activity? Some authors in the literature mention that the gummies present a water activity much lower than the one reported, justify their results amply by comparing them with the bibliography.

Lina 102. Prediction models should include the R2 coefficient. Are the colors used in the 3D plots statistically different? The values of springiness, cohesiveness and gumminess seem not to be close to the statistical model. It would be advisable to formally present the result from the predicted value to the observed value.

The authors should include the statistical prediction tables of the mathematical models including the F coefficients and the significance factor. The authors should present the results of normality and kurtuosity of the experimental data.

Line 148-151. The authors mention that ". In this way, despite having completely replaced sucrose and glucose with agavins and agave syrup, it was observed that these functional ingredients did not alter the textural properties of the gelatin gummies." They should reconsider this sentence, since the statistical model shows that the use of agavins and agave change the textural properties of the gummies.

It is very confusing that the axes of the 3D graphs change between agavin and agave whey. It would be highly recommended to edit them to keep the agavin content on the x-axis and the syrup content on the y-axis, as described in the equations.

Table 2.

Presenting the moisture data of the GC samples. Greater than 20% is very ambiguous, for example the reader might assume 95% moisture.

Most of the data presented in Table 2 are very different from those associated with the optimal gummi. If the selected statistical desirability was greater than 0.9, why are these values so different? Did the model not work?

Line 207. Authors results and discussion with similar works such as Rodriguez-Rodriguez, R.; et al (2022). https:// doi.org/10.3390/molecules27154902

Line 215-218. The authors indicate that 5 images were used for the estimation of cell viability. Is the number of images statistically significant, what number of cells was estimated with 5 images, could viability be influenced by the section of the gummy used, were the images taken from different gummies at each time interval, and were the images taken from different gummies at each time interval?

After day 14, 2the probiotics are viable, but do not grow", what does this mean? The authors should explain this behavior.

What percentage of variability is explained by the PC1 and PC2 components in the principal component analysis? Is it sufficient with these two components? Is it representative to use 5 samples of the same gummy for this study?

Methodology section

Line 294. What is the solids concentration of the ingredients agavinas and agave whey? Due to the high temperatures associated with the cooking process, the authors should include evidence of the degree of polymerization and carbohydrate profile presented in the samples.

Line 314. A wide variety of products with these characteristics are available on the market. The authors should describe the desirability properties considered.

The authors do not present the number of samples used for statistical measurements. It would be advisable for the audience to present the comparative TPA analysis graphs between the prebiotic sample and the symbiotic sample.

Reviewer 2 Report

Comments and Suggestions for Authors

In this study, the authors encapsulated S. boulardii in alginate microcapsules, which were then added to an optimized gummy formulation to create synbiotic gummies. These synbiotic gummies were evaluated for various parameters such as texture, water activity (Aw), humidity, and sugar content. The study also assessed the probiotic viability and changes in the gel structure of the microcapsules over 24 days.

The study is timely and relevant to the food industry. However, there are certain concerns, as mentioned below.

L25-26: Please say more about the experimental design, variable ranges, and responses analyzed.

L37-74: Please make it crisp.

L88: The “synbiotic” characteristics, especially the prebiotic activity, have not been evaluated.

L102: What are x and y? Are they coded form or real form? What about the ANOVA data for the model and p-value for the individual term? Are all the terms in Eq 1 to 6 significant?

L116: Please cite suitable references.

L125: Whether the interaction effect was synergistic or antagonistic? Please discuss this from that perspective. It is better to present the regression coefficients in the model in coded form.

L140: Check the sentence formation.

L158: What is the formula for global desirability employed here?

In Table 1: How were ‘s’ or ‘t’ values selected?

Table 2: What about comparing gummies using conventional techniques or pure sucrose?

L181: Please check the sentence.

L192: What was the reason for this?

L204: Should it be during consumption or before storage?

L248: Not clear. Please provide more clarity.

L269-271: Please add suitable references.

L300: Why RCCD design was used. A mixture design is recommended to optimize any formulation. For each run of Table 3, the summation of Agave syrup (mL)+ Agavins (g) is different. How was the total mixture mass kept fixed for comparison?

L308: Just saying RSM was employed is not enough. Please elaborate on model development, model equations, optimization criteria, weightage fixed for responses, and objective function based on which the optimization was conducted.

L353: What was the microbial population in this concentration?

L372: Why 50C? Is there any reason for justification?

L377-378: Please rewrite accordingly. The meaning is not clear.

Reviewer 3 Report

Comments and Suggestions for Authors

Review on manuscript: gels-2946835:

Effect of agavins and agave syrup use in the formulation of a synbiotic gelatin gummy with microcapsules of Saccharomyces boulardii

by Liliana K. Vigil-Cuate, Sandra V. Avila-Reyes, Brenda H. Camacho-Díaz, Humberto Hernández-Sánchez, Perla Osorio-Díaz, Antonio R. Jiménez-Aparicio, Paz Robert, Martha L. Arenas-Ocampo

submitted to Gels

In the manuscript submitted for comments the Author studied the effect of agavins and agave syrup on some properties of a synbiotic gelatin gummy with microcapsules of Saccharomyces boulardii.

In my opinion, the manuscript is interesting and the solution proposed by the authors may be applicable in practice, but it is a pity that the authors did not use sensory analysis as one of the research methods.

Detailed recommendation:

line 105 – in the materials section, the authors do not mention the use of commercial gummies, what is their origin and basic composition?

line 140 – style should be corrected,

Table 1 – combining a chart with a table is not a good solution,

line 173 – a better solution is to explain the abbreviation under the table and not refer the reader to other parts of the manuscript,

line 181 – style should be corrected,

line 190 – superscript should be used,

lines 190-191 – log and Log records should be standardized,

line 210 – superscript should be used,

lines 295-296 – some basic information about the characteristics of agavins and agave syrup should be provided, e.g. water content, extract content, sugars content, etc. some basic information about the characteristics of agavins and agave syrup should be provided, e.g. water content, extract content, sugars content, etc.,

lines 318-319 – tap or deionized water?

line 365 – subscript should be used,

line 369 – isn't this methodology given in section 4.3?

line 377 – what color analysis do the authors have in mind? where is the description of the method and the results obtained?

line 378 – isn't this methodology given in section 4.4?

References – the use of full titles or abbreviations of journal titles should be standardized.

Round 2

Reviewer 1 Report

Comments and Suggestions for Authors

Line 44, delete the accent.

Line 45- The authors mention "Agavins (agave fructans), are low molecular weight carbohydrates," however there are reports where the dP can reach more than 50 fructose units. Improve the wording.

In the objective, the authors mention "agave syrup as a sweetener". No measurements of the degree of sweetness were presented. It is suggested to rewrite the objective as close as possible to its scope.

Reviewer 2 Report

Comments and Suggestions for Authors

The authors have made necessary changes as per the queries or concerns raised by the reviewer. 
